**METHOD**                                                                      **Open Access**

# TAD fusion score: discovery and ranking the contribution of deletions to genome structure

Linh Huynh[1] and Fereydoun Hormozdiari[1,2,3*] ⓘD

**Abstract**

Deletions that fuse two adjacent topologically associating domains (TADs) can cause severe developmental disorders. We provide a formal method to quantify deletions based on their potential disruption of the three-dimensional genome structure, denoted as the TAD fusion score. Furthermore, we show that deletions that cause TAD fusion are rare and under negative selection in the general population. Finally, we show that our method correctly gives higher scores to deletions reported to cause various disorders, including developmental disorders and cancer, in comparison to the deletions reported in the 1000 Genomes Project. The TAD fusion score tool is publicly available at https://github.com/HormozdiariLab/TAD-fusion-score.

## Introduction

Eukaryotic genomes consist of multiple chromosomes, each chromosome is a linear sequence. Genes and other regulatory elements are organized and positioned in this linear sequence. However, in reality, chromosomes are folded into a complex three-dimensional (3D) structure. This 3D structure brings distal genomic elements into direct contact to make them interact with each other. The interaction between functional elements such as enhancers and promoters plays an important role in controlling biological processes such as transcription, replication, and DNA damage repair. [1, 2].

The first in-depth studies on the 3D structure of chromosomes were obtained using microscopy techniques [3]. However, these approaches tend not to be able to resolve the physical interactions and the 3D structure in high-resolution and high-throughput fashion [4, 5]. With the advent of various biomolecular chromosome conformation capture (3C) techniques in the past few years, our understanding of the 3D genome structure and its contribution to various biological processes has been revolutionized [4–7].

Recently, an extension of the 3C technique has been developed for analyzing the genome-wide interaction of all chromosomes in high-throughput and high-resolution fashion denoted as Hi-C [8–11]. Hi-C technique that is based on proximity ligation and pair-end sequencing allows researchers to capture the 3D structure of a chromosome at a kilobase pair (kbp) scale. This technique produces a genome-wide sequencing library to construct a contact frequency matrix that provides a proxy for measuring the three-dimensional distances among all possible locus pairs in the genome [12, 13].

One of the most novel discoveries using Hi-C data is the partition of the genome into hundreds of kilobase pair segments which interactions are highly enriched inside each segment and are significantly depleted between adjacent segments. These segments are denoted as topologically associating domains (TADs) and can be seen as continuous square domains on the diagonal of the Hi-C contact matrix [2]. It is shown that TAD boundaries are significantly enriched with insulator proteins such as CTCF in mammalian cells [1, 2, 14, 15]. These proteins are shown to be able to block interactions between different regulatory elements and are the main reason why there is significant depletion of interactions between two adjacent TADs. TADs are hypothesized to be conserved between different cell types and across close species. However, it is not trivial to quantify this mainly due to difficulties

*Correspondence: fhormozd@ucdavis.edu
[1]Genome Center, UC Davis, Davis, USA
[2]UC Davis MIND Institute, Sacramento, USA
Full list of author information is available at the end of the article

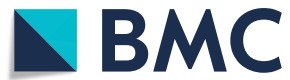

of accurate TAD discovery, existence of sub-TADs and nested TADs [16–21].

In a recent experiment, Dali and Blanchette [19] manually annotated TAD structures in a randomly selected segments of genome and compared against TAD prediction using various computational tools. They reported a significant discordance between various prediction algorithms. Furthermore, it was observed that tools generally had low sensitivity, often picking up less than 10% of manually annotated TAD structures. In fact, almost 25% of the manually annotated boundaries never got detected by any of the tools [19]. In other studies [22, 23], a similar disagreement between computational tools for TAD prediction was observed. Existence of sub-TADs, nested TADs [18–20, 24], and limiting assumptions made by different computational methods are some of the main reasons for these discrepancies. It was also argued that parameter selection, normalization, and matrix correction can have a significant impact on the final result of these tools [23]. Although the number and the size of TADs vary significantly among different tools, this result is still more comparable than the one from determining specific loops by different tools [17, 23].

In a seminal paper by Lupiáñez et al. [25], the authors showed that structural variations (SVs) that disrupted TADs (by deleting TAD boundaries) can result in novel enhancer-promoter interactions which in return can cause severe developmental disorders [26, 27]. More specifically, it was shown that structural variations that disrupted the TAD boundary of WNT6/IHH/EPHA4/PAX3 locus fused two TADs which caused malformation syndromes [25]. Similar findings have shown that structural variations (SVs) that cause TAD organization disruption and fusion of TADs at different loci can cause various developmental disorders [25, 28–34]. A recent study [35] also reported that short tandem repeats related to diseases also co-localized at TAD boundaries. In addition to the developmental disorder, recent findings also reported similar TAD disruptions in cancer cells [36–39] as an effective mechanism for the activation of oncogenes. For example, it was shown that deletion at the TAD boundary resulted in an activation of oncogene *TAL1* [36]. In another example, a tandem duplication at the TAD boundary also resulted in activating a gene locus relevant to cancer *IGF2* [40].

These studies have shown the importance of being able to study the impact of structural variations on the three-dimensional genome organization. Specifically, if a deletion results in the disruption of two or more TADs by fusing them and creating novel genomic interactions (Fig. 1). However, no method exists for scoring and ranking deletions by predicting explicitly their effect on the 3D genome structure. For example, Li et al. [41] introduced the Structure Influence score but their method did

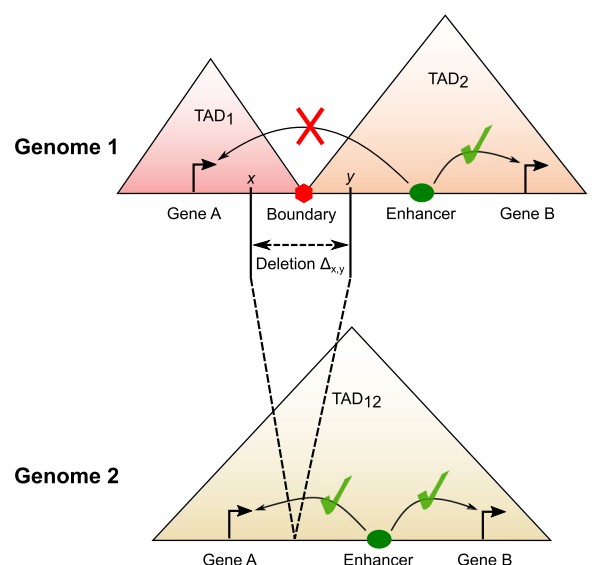

**Fig. 1** An illustration of the TAD fusion where a deletion can affect the genome structure. The original genome (upper) has two TADs separating by a TAD boundary. Any locus of one TAD can interact with other loci in the same TAD (e.g., the enhancer and gene B of $TAD_2$). But a locus cannot interact with any other locus of the other TAD (e.g., gene A of $TAD_1$ and the enhancer of $TAD_2$). In the mutant genome (lower), two original TADs are fused into one TAD since the TAD boundary is removed by the deletion. Thus, any two loci in the fusion TAD can interact although they do not interact in the original genome (e.g., the enhancer and gene A)

not show specifically how chromatin interaction might be changed due to a deletion. In this paper, we provide a formal method to scoring deletions based on predicting their effect on the three-dimensional genome structure, denoted as TAD fusion score. We validate our score with deletions that were previously tested (e.g., Sox9-Kcnj2, HoxD, and Firre deletions). Furthermore, we show that deletions that cause a TAD fusion are rare and are negatively selected against in a general population. Finally, we show that our method correctly gives significantly higher scores to deletions that cause various disorders (developmental disorders and cancer) in comparison to deletions reported in the 1000 genomes project.

## Results
### Methods' overview
To study deletions and their contribution to disease and evolution, we define a score for any deletion based on its level of modifying the 3D genome structure and potential of fusing two adjacent TADs. We call that score as *the TAD fusion score* and develop a novel computational method to calculate it in the "Methods" section. In this paper, we present the method and the analysis result of the deletion, but similar ideas can be extended to consider other SVs such as inversions or translocations (Additional file 1: Figure S1).

Formally, we define the TAD fusion score as *the expected total number of changes in pairwise genomic interactions* as a result of a deletion. Thus, the input of our computational method consists of the Hi-C matrix of a genome with a reference allele (i.e., without the deletion) and the coordinates of a deletion of interest. The main output of the method is a score representing the expected number of changes in the pairwise genomic interactions as a result of the deletion.

We propose a *two-step framework* for calculating the TAD fusion score: (i) predicting a new Hi-C contact matrix $G$ of the mutated genome (i.e., with the deletion), given as input the Hi-C contact matrix $H$ of a genome without the deletion and the deletion coordinates; (ii) comparing the predicted Hi-C contact matrix $G$ with the modeled Hi-C contact matrix $H'$ (as the by-product of step (i), see the "Methods" section) to estimate the number of changes in pairwise genomic interactions as a result of the deletion.

### Data

For TAD fusion score analysis, we used 5 kbp resolution Hi-C data of the human cell GM12878 from in situ experiments [42]. We chose the 5 kbp resolution since the TAD structure conservation (with other cell lines) was validated at that resolution. For deletions, we used the reported deletions in the 1000 genomes project (1KG) [43], the fixed deletions in great ape lineage [44], the deletions reported in tumor samples from The Cancer Genome Atlas (TCGA https://cancergenome.nih.gov/) project and a small set of deletions reported and validated to cause developmental diseases by disrupting the 3D genome structure [25, 28–30] (Additional file 1: Table S1).

For baseline methods, we predicted the TAD fusion by using the TAD boundaries predicted by several representative TAD callers [42, 45, 46] and the insulation score from [47]. The setting of these methods is presented in the supplementary information (Additional file 1: Table S2). For validating the overlap of the predicted insulation with reported CTCF binding sites, we used the CTCF peak data from ENCODE [48]. All data (Hi-C, 1KG deletions, deletions of GM12878, CTCF binding sites) were aligned with the reference human genome b37 (hg19). Note that 82 deletions reported in sample GM12878 were excluded from the 1KG deletion set as they were also in the reference Hi-C sample. Since we used Hi-C data at the 5 kbp resolution, only deletions that were longer than 10 kbp were considered to ensure each deletion removed completely at least one 5 kbp-bin. There was a total of 7383 such deletions reported in the 1000 genomes project [43] satisfying the above criteria. The full list of datasets used for this study was summarized in the supplementary information (Additional file 1: Table S3).

### Evaluating the model and the prediction

We first evaluated the capability of our proposed model (using bin pairs with distances at most 2 Mbp for training) in producing the approximate Hi-C data matrix using the Hi-C datasets (of chromosome 1) of seven different cell lines from [49] as input data (see Additional file 1: Table S4). We compared raw Hi-C matrices (denoted as *Raw*) and the Hi-C matrices produced using our proposed method (denoted as *Model*). The stratum-adjusted correlation coefficient (SCC) of the HiCRep package [50] was used to estimate the similarity between any pair of Hi-C matrices (i.e., *Raw* versus *Model*). We observed that the SCC score between a raw Hi-C matrix and our modeled Hi-C matrix was very high for all the seven matching raw versus model pairs (ranging from 0.948 to 0.976, see Additional file 1: Table S4). More importantly, our modeled Hi-C matrix of each cell line was closest (highest SSC score) to the raw Hi-C matrix of the same cell line (the bold values on the diagonal in Additional file 1: Table S4). Conversely, the raw Hi-C matrix was also the closest one to the modeled Hi-C matrix of the same cell line. This indicated that our method was capable of accurately capturing the subtle difference between cell lines from their Hi-C data matrices. Furthermore, we also examined further two pairs of closely related cell lines H1 vs ME and MS vs IMR90 (based on results and data from [51] which includes 40 kbp resolution Hi-C data). As depicted in Fig. 2a, our estimated Hi-C matrix (model) was the closest one to the raw Hi-C matrix of the same cell type based on the SCC score. Conversely, the raw Hi-C matrix was also the closest one to the estimated Hi-C matrix (our model) of the same cell type. We also observed that our model reduced the difference between two cell lines in comparison to the raw Hi-C data (i.e., two estimated matrices were closer to each other than two raw ones were). This was expected since our approximation formula only took into account the length and the insulation while other details such as the loop of some specific loci (i.e., "peak" as in [42]) were not utilized.

Second, we also evaluated if the bins that had large insulation predicted by our method were enriched with CTCF, as it was widely reported that the TAD boundaries were enriched with CTCF binding sites [2]. We compared our predicted insulation at each bin (i.e., parameter $r_i$, using bin pairs with distances at most 400 kbp for training, see the "Methods" section) with the CTCF peak at that bin for three different cell lines H1, IMR90, and GM12878 (where their CTCF data was available from ENCODE [48]). As expected, we did observe a significant enrichment of CTCF (by the average number of peaks) at bins predicted to have a higher insulation (parameter $r_i$) by our proposed model (Fig. 2b). Furthermore, this observed enrichment held for all three cell lines and Hi-C matrices tested.

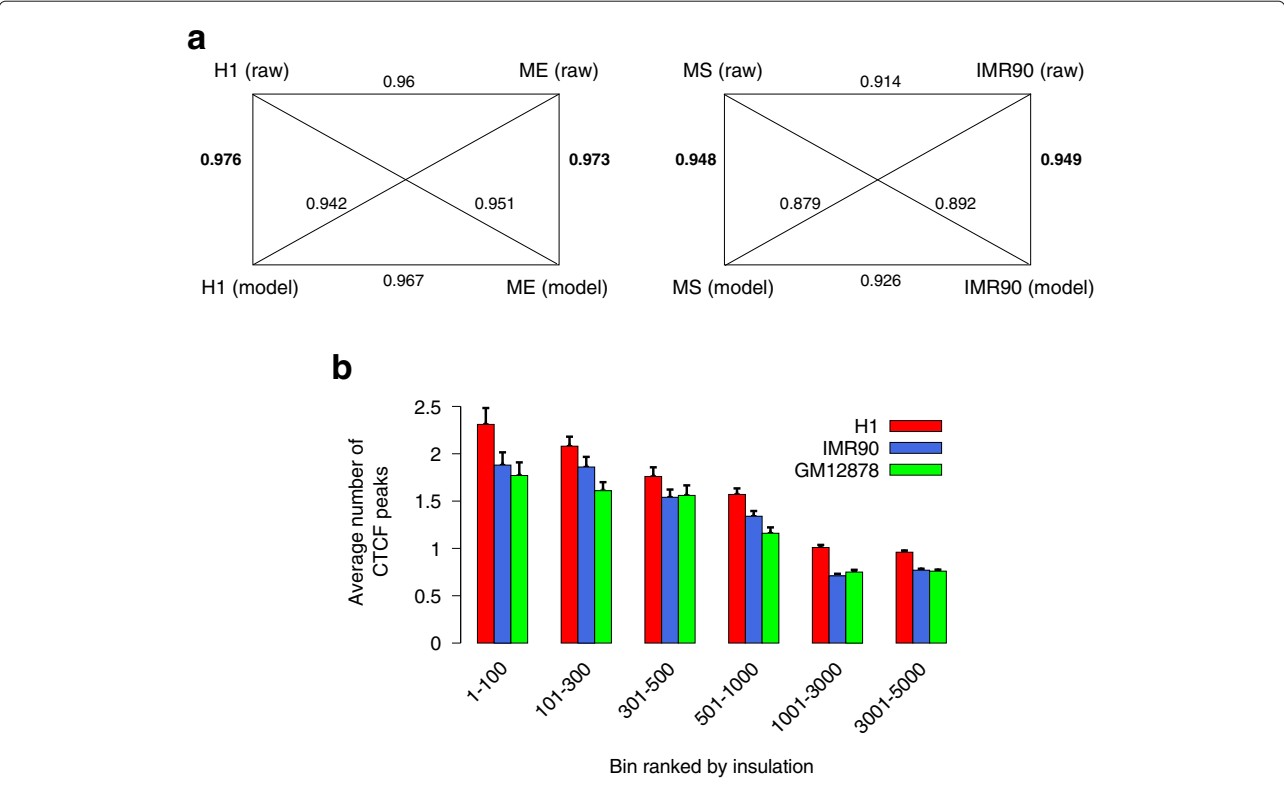

**Fig. 2** Validation of our model. **a** Comparison between our modeled Hi-C matrices and raw Hi-C matrices between closely related cell types; the numbers on edges represent the SCC score (from HiCRep [50]). **b** Enrichment of CTCF at bins that have a high insulation predicted by our model from Hi-C datasets [49]. In both **a** and **b**, Hi-C data is of chromosome 1

Third, we also compared the predicted insulation at each bin (i.e., parameter $r_i$, using bin pairs with distances at most 400 kbp for training as above) for seven different cell lines from [51] (i.e., H1, ME, TB, NP, MS, IMR90 and GM12878). We built a pairwise similarity matrix for these seven cell lines using the insulation at each bin (Additional file 1: Figure S2). The hierarchical clustering tree built from this similarity matrix was comparable to the one built using A/B compartments reported in [51] using the same set of cell lines (with one minor exception of swap in the order of TB and NP). Furthermore, the hierarchical clustering produced by the similarity matrix was in a complete concordance with the clustering reported in [49]. Other important features were similar to previous studies [49, 51, 52] such as (H1, ME) and (MS, IMR90) were grouped as pairs while GM12878 was separated as an outbranch in the tree. This experiment indicated that the insulation calculated using our method was capable of capturing the subtle difference between cell types and correctly grouped more closely related cell types together (e.g., H1 and ME).

Finally, we investigated the performance of our proposed approach for predicting Hi-C contact frequencies after deletions against the experimental Hi-C data for the same set of deletions. We used deletions at Sox9-Kcnj2,

HoxD, and Firre regions where Hi-C data existed for this comparison [34, 53, 54]. We compared the predicted Hi-C data and the experimentally measured Hi-C data (see Additional file 1: Figures S3-S4). Since these interactions were at variable distances, again, we used the stratum-adjusted correlation coefficient (SCC) from HiCRep package [50] to evaluate the similarity instead of the regular Pearson correlation coefficient. For the three deletions (at Sox9-Kcnj2, HoxD and Firre), our predicted Hi-C data was quite similar to the experimental Hi-C data of these deletions with an average SCC of 0.82. Interestingly, in the case of the deletion at the Sox9-Kcnj2 boundary, we observed that there was a very small TAD (or sub-TAD, on the downstream of the deletion) that was fused to a large TAD on the upstream of the deletion (as can be seen using both our predicted Hi-C matrix and the real experimental Hi-C matrix in Additional file 1: Figure S3). This fusion might cause the increase of Kcnj2 expression reported in [34].

### Evaluating the TAD fusion score
As demonstrated, our proposed model is accurate in predicting the Hi-C changes due to a deletion (Additional file 1: Figures S3-S4). We also compared our proposed TAD fusion score in quantifying the contribution

of a deletion in altering the 3D genome structure versus the prediction resulted from utilizing the state-of-the-art TAD and insulation callers. We applied our TAD fusion scoring method (see the "Methods" section) for evaluating reported deletions in the 1000 genomes project (1KG) using the Hi-C dataset of GM12878 at 5 kbp resolution [42]. We compared our TAD fusion score versus two different general approaches (i) TAD prediction methods (Arrowhead [42], Insulation Score [46], and CaTCH [45]) and (ii) an insulation score calculated per bin using LRI score [47].

The first comparison was against different methods for TAD prediction [42, 45, 46]. We observed that the predicted TAD boundaries varied significantly between different TAD callers and there was only a small fraction of boundaries predicted by all the tools (Additional file 1: Figure S5). Interestingly, we observed that the average TAD fusion score calculated by our approach was higher for deletions that were predicted to remove a TAD boundary based on the majority of the tested TAD callers (Additional file 1: Figure S5).

The second comparison was against the approach that used the maximum LRI score (i.e., an insulation score predicted by [47]) of deleted bins to rank or score each deletion. We did observe a significant yet moderate correlation between the TAD fusion score assigned to each deletion and the maximum LRI score assigned to any bin inside that deletion ($r = 0.43, p < 1e − 10$, for chromosome 1). Further investigation has revealed that some of the discordance between the TAD fusion score of a deletion and the maximum insulation score (LRI) inside this deletion came from the cases where there were more than one strong insulators near the TAD boundary. In these cases, some of insulators were not deleted as a result of the deletion (see Additional file 1: Figure S6, visualized with Juciebox [55]). Thus, the deletion of a bin with a high insulation score might not result in a TAD fusion in these cases. Our method overcomes this limitation by taking into account both the deleted insulators and the remaining (i.e., not deleted) insulators instead of considering only deleted ones.

### TAD fusion is under negative selection
In few recent studies, it was shown that deletions that disrupted the genome structure and caused a TAD fusion could result in various developmental disorders. Thus, it was hypothesized that TAD fusion events should be negatively selected against during evolution [56, 57]. In a recent comparative study on the gibbon genome against the human genome it was shown that structural variations tended to avoid disrupting the TAD structure [56]. It was suggested that the increased selective pressure against SVs that disrupted TADs was the main cause [56]. Furthermore, in another recent study, it was shown

that significantly lower percentage of fixed deletions in great ape [44] removed a TAD boundary from what was expected by chance [57]. We also tested this hypothesis using our method for TAD fusion scoring and ranking. We compared the TAD fusion scores calculated for non-disease deletions observed in comparison to a null model where deletions of the same length were randomly assigned in the genome. We used two different deletion sets for this experiment: (1) the reported fixed deletions in great ape lineages [44] and (2) deletions reported in the 1000 genomes project [43]. We randomly permuted the deletions from both sets to ensure the same number and length of deletions per chromosome and calculated the TAD fusion score per permuted deletion for each set and repeated this permutation procedure for 1000 times. We then compared the number of deletions in great ape lineages and deletions in 1000 genomes project with TAD fusion score above various cutoffs (Fig. 3a and b) against random permutation sets (i.e., the null model). We observed the number of deletions with a high TAD fusion score was significantly less in the great ape lineage set and the 1000 genomes project set in comparison to the set of randomly permuted deletions (for all the tested cutoffs, we got a significant $p$ value $< 1e − 15$, Fig. 3a and b). The combination of the great ape deletions and the 1000 genomes deletions supported the hypothesis that deletions that caused TAD fusion were selected against in the evolution. Finally, we investigated the correlation between the allele count of reported deletions in the 1000 genomes project against the calculated TAD fusion score. We observed that the allele counts were inversely correlated ($\rho = −0.11, p < 1e − 15$) with the TAD fusion scores (Fig. 3c). Taking together, these data and analysis supported the hypothesis that deletions with a higher TAD fusion score were under a stronger selection.

### Contribution of TAD fusion to diseases
To study the contribution of TAD fusion to human diseases, we utilized a list of eight deletions that were validated to cause various disorders as a result of the genome structure disruption [25, 28–30] (Additional file 1: Table S1). We calculated the TAD fusion score of these eight deletions and compared against the TAD fusion score of reported deletions in the 1000 genomes project. The result showed that the TAD fusion score of these disease deletions was significantly higher than the one of almost all 1KG deletions (Fig. 4a). Note that, there was an expected correlation between TAD fusion score and the length of the deletion since (i) longer deletions would result in two loci which were much farther apart to get adjacent and cause a 3D structure modification and (ii), more importantly, a longer deletion had a higher chance of deleting a TAD boundary. Thus, we tried to further investigate the TAD fusion score of these disease deletions with 1KG

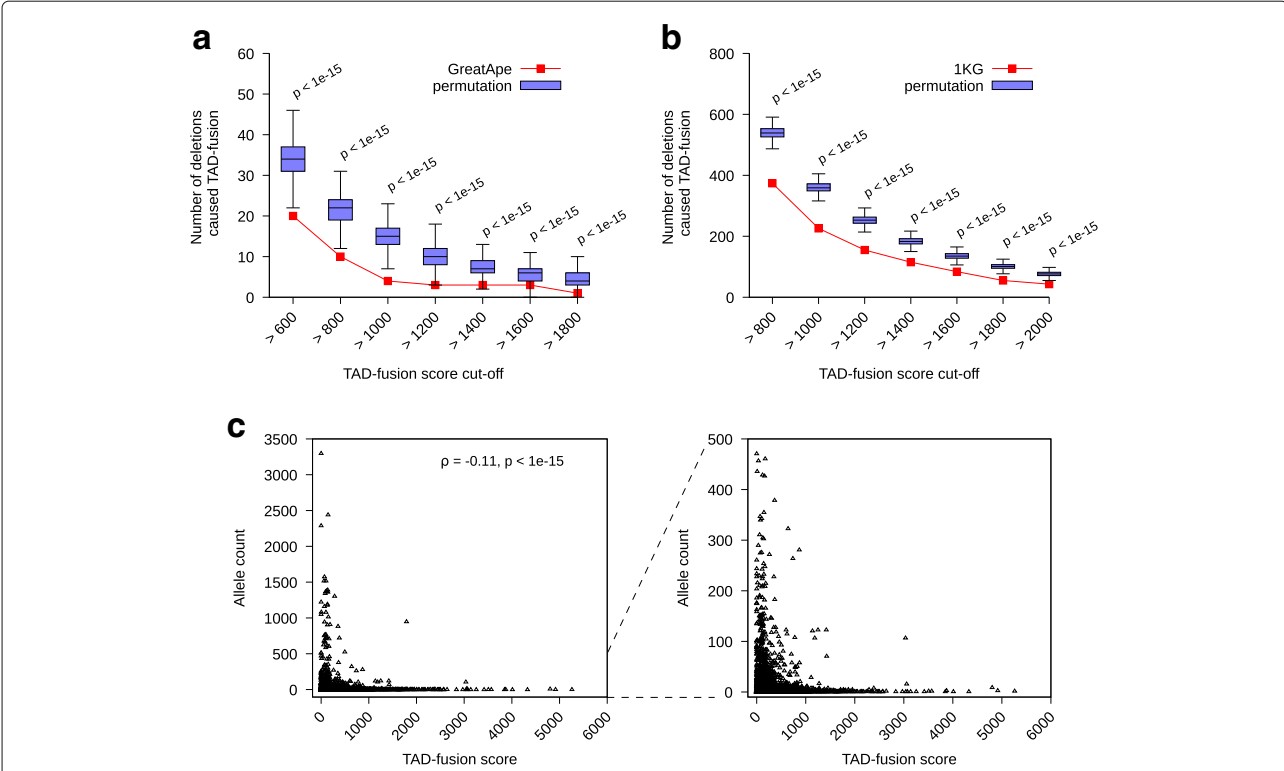

**Fig. 3** TAD fusion score of the deletion in the 1000 genomes project (1KG) and fixed great ape deletion sets. **a** The number of deletions with high TAD fusion score from the fixed great ape deletion (red curve) and (1000 random) permuted data (blue boxplots) for various cutoffs of TAD fusion score. **b** The number of deletions with high TAD fusion score from the 1KG data (red curve) and (1000 random) permuted data (blue boxplots) for various cutoffs of TAD fusion score. **c** The inverse correlation of allele count of 1KG deletions with the TAD fusion score

deletions of the same length range. However, a direct comparison of disease deletions with 1KG deletions of the same length was not feasible due to a significant difference between the length of disease deletions versus the length of 1KG deletions. However, for four of these disease deletions, we were able to infer short deletions which excluded regions which overlapped other deletions known to not cause the disease. For example, Lupiáñez et al. [25] reported a deletion DelB (chr2:221278232–223014332, hg19) that caused the limb malformation disease. This deletion was over 1.3Mb long, and they also confirmed an equivalent deletion in mice (chr1:76388978–78060839, mm9) that caused this disease. However, they also validated a control deletion at chr1:76388978–77858974 and they confirmed that mice carrying this deletion did not show any abnormality in limb development [25]. Therefore, we inferred that the smaller deletion chr1:77858974–78060839 might be the cause of the limb malformation (the remaining part of the disease deletion after excluding the deletion which did not cause the disease). We lifted over this mm9 coordinates to get the hg19 coordinates of chr2:222800125–223011646. This inferred deletion was only 212 kbp, and it can be compared with some 1KG deletions at the same length range (see Fig. 4b). We inferred four such deletions (see Additional file 1:

Table S5). We compared these deletions with 1KG deletions at the same length range. We observed a significant increase of the TAD fusion score of inferred disease deletions against the TAD fusion score of 1KG deletions of the same length range (Fig. 4b). This supported the claim that our method assigned higher scores to deletions which disrupted the genome structure and caused the TAD fusion.

We also compared the TAD fusion score of fixed deletions in great ape lineages [44], 1KG deletions, and TCGA (cancer patients) deletions at the same length ranges. We limited the deletions from TCGA data to an upper bound of 500 kbp to avoid the bias of very long deletions. A recent study reported the negative selection of deletions of TAD boundaries in a pan-cancer analysis [52]. Furthermore, they showed that not only very strong TAD boundaries were protected from the deletion, but they tended to be co-duplicated with super enhancers [52]. In this study, we explored further the TAD fusion score of deletions reported in tumor cells from TCGA (The Cancer Genome Atlas) by comparing it against the TAD fusion score of 1KG deletions of similar length ranges (Fig. 4c). We observed that for most of deletion length ranges, the TCGA deletions had a significantly higher TAD fusion score than 1KG deletions (Fig. 4c). Interestingly, the

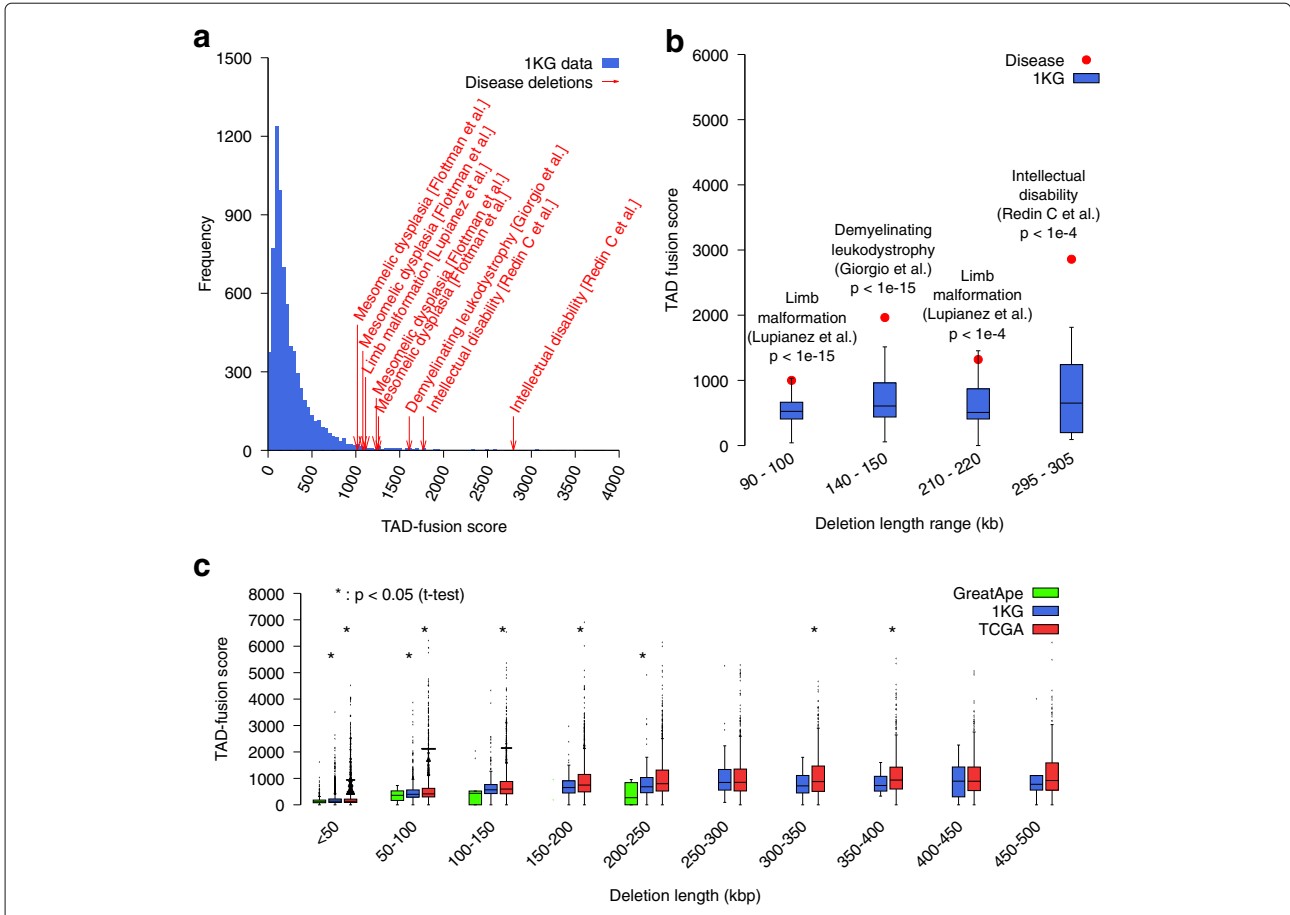

**Fig. 4 a** The score of 1KG deletions and eight reported disease deletions that cause disorders by disrupting TAD structure (i.e., TAD fusion). **b** Comparison between the TAD fusion score of inferred deletions and the TAD fusion score of 1KG deletions at the same length range. **c** Comparison of TAD fusion score between deletions observed as fixed in great ape lineages, 1KG (healthy samples) deletions, and cancer (TCGA) deletions. In **c**, the notations (*) at lower and upper positions represent the comparison between great ape lineages vs 1KG and 1KG vs TCGA respectively

average TAD fusion score was lowest for the fixed deletions in great ape lineages, followed by 1KG deletions and was highest for the TCGA (cancer) deletions (Fig. 4c). On the one hand, this indicated that the oldest set of deletions (i.e., the fixed deletion in great ape lineages) had gotten the lowest TAD fusion score. On the other hand, the deletions which almost had not been evolutionarily selected against (the somatic deletion in TCGA) had gotten the highest TAD fusion score.

## Discussion

Chromosome structure folding in the three-dimensional space plays an important role in regulating gene expression. Hi-C data provides an experimental evidence that the chromosome is organized into modular hierarchical domains such as compartments, TADs, and sub-TADs. Motivated by few case studies of the limb development and the developmental disability, it is hypothesized that deletions (and other SVs) can cause 3D chromosome structure disruption and contribute to the etiology of

these disorders. In most cases, these deletions contribute to the disorder by fusing two TADs. Thus, these types of deletions should be negatively selected for and they should be rare in normal samples. Recent studies [56, 57] also reported this negative selection but they used the predicted TAD boundary that varied by different heuristics of different computational methods [17, 19]. In this study, we have developed a novel computational method for assigning TAD fusion score to any input deletion based on its potential contribution to fusing TADs. As part of this method, we solve two related problems: (i) The first problem is how we can predict the changes to the Hi-C matrix due to a deletion in the genome, (ii) and the second problem is how we can compare two Hi-C matrices to find the significant difference between them.

For the first problem of predicting the changes in the Hi-C matrix due to a deletion, a more accurate model is the key to improve the prediction. Recent work [58] proposed a polymer model to predict the Hi-C data change due to SVs, but it does not take into account the loop extrusion

mechanism that is supported directly by recently experimental evidence [59]. In addition, the prediction in [58] is based on an intensive simulation that is not applicable for analyzing a large number of deletions as we did for the 1KG and its permutation dataset. Our succinct model presented here agrees well with the experimental data by taking into account the power law and the insulator that are based on the random diffusion mechanism [60] and the loop extrusion mechanism [61, 62] respectively. It is widely accepted that CTCF plays an important role in forming the TAD structure [63–65]. However, only 15% CTCF sites are at the TAD boundary while 85% CTCF sites are inside the TAD [63]. We still have not understood the underlying mechanism to distinguish the role of each site (based solely on the binding strength from ChIP-seq data) in forming the 3D chromosome structure. Therefore, our current method does not use the CTCF binding data to fit the model, but we can extend the method to utilize that data with our further understanding of the CTCF role in the future. A recent study [66] proposed a trick to approximate the physical distance from the shortest path method. It is also interesting to evaluate if this trick is applicable to improve the prediction. In our algorithm, we assume that the deletion only changes the interactions that cross the deletion (i.e., between two bins, one is in the upstream and one is in the downstream of the deletion). However, this constraint may not always hold and the model should be extended to consider also the changes between bins at one side of the deletion. Furthermore, the model could also take into account other biological features such as histone modification marks, DNA methylation, or DNA accessibility [45, 67].

For the second problem, we can evaluate the change of only enhancer-promoter interactions rather than the interaction between all bins. In addition, the comparison on the contact frequency should also be normalized for each specific gene or region to achieve a better result. It is important to note that although the TAD boundaries are believed to be conserved in most cell types, there still might be important differences in the genome structure between different cell types or different development stages. This difference should be reflected in the Hi-C matrix of that cell type. Furthermore, while one deletion might not cause a significant genome structure disruption in one cell type, it might do so in another cell type. Thus, as more high-quality Hi-C data of different cell types are produced, the deletions should be tested and scored using each of the Hi-C matrices. By that, we can maximize the possibility of finding deletions which might cause TAD fusion in a specific cell type or a specific development stage.

Further experiments are still needed to validate our predicted results. For example, we can make a deletion in the genome (e.g., with CRISPR) and generate the new Hi-C data and also measure the gene expression change. The gene expression may not change constantly since the regulation depends not only on the genomic interactions but also on other factors (e.g., the presence or the absence of transcription factors); thus, we may need to measure the gene expression at different cell states for the validation. However, this obstacle could be overcome since more data are contributed by the community [53, 54].

In addition, due to the complexity of the original optimization problem, we have applied the approximation for some steps, any improvement on this approximation to make it closer to the original one may improve the result. Since the LP solver still has a large complexity, any improvement on reducing the problem size is necessary. The number of variables can be reduced by simplifying the objective function such as removing less significant location pairs while adding larger weights for important ones. Other metrics rather than L1 in the objective function may also need to be evaluated. Further, the model is fitted in the log scale, and thus, zero entries should be normalized; this problem is similar to the problem of normalizing drop-out data in processing the single-cell gene expression data. In this study, all deletions that are shorter than 10 kbps are skipped due to the resolution 5 kbps of the Hi-C data. But this limitation can be overcome as more high-resolution Hi-C data will be contributed by the community [68] and the computational tool such as [69] can also help us predict the high-resolution data.

There are several applications of the proposed method for TAD fusion discovery, it will provide biologists a way to rank and pick deletions that potentially cause a significant disruption on the genome structure. Furthermore, TAD fusion discovery will also provide a novel mechanistic explanation of how a group of non-coding deletions (and SVs in general) is contributing to developmental disorders or cancer. Finally, the approach presented here for deletions can be extended to consider other types of structural variants, such as inversions and translocations (see Additional file 1: Figure S1).

## Methods
Our goal is to develop a *computational method to provide a score to a deletion mutation based on its level of modifying the 3D genomic structure and potential of causing a TAD fusion.* We are assuming the input to the method consists of the Hi-C matrix of the genome with reference allele (i.e., without deletion) and the coordinates of the deletion, and the output is a score representing *the number of new genomic interactions* made (i.e., TAD fusion score) as a result of the deletion. For this paper, deletions are assuming only homozygous and non-overlapping.

## Definitions and notations:

- Chromosome location bin: Achieving single base-pair resolution is impossible due to the limited sequencing coverage that produces the Hi-C data. Thus, it is common in practice to partition the genome of a chromosome into non-overlapping segments (bins) of the same length (e.g., 5 kb) to summarize the data. For a chromosome, we assume that we have $n$ bins which are numbered across the chromosome as $1, 2, 3, \ldots, n$.

- Hi-C contact frequency map/matrix: Let $H \in R^{n \times n}$ be a symmetric matrix constructed from Hi-C data where $H_{i,j}$ is the average contact frequency between $i^{th}$ bin and $j^{th}$ bin (i.e., $H_{i,j}$ is large if $i$ and $j$ are close in 3D space, otherwise, $H_{i,j}$ is small if they are not close in 3D space). Note that, if two bins $i$ and $j$ are interacting with each other, then they have to be very close in 3D space.

- Deletion structural variation: We denote $\Delta_{x,y}$ as the deletion from the $x^{th}$ bin to the $y^{th}$ bin of a chromosome. We round the coordinates down to calculate the location bin of a deletion.

- Genomic interaction: Two bins $i$ and $j$ are considered interacting if they create a physical interaction in three-dimensional space. Some of the well-known examples of such interactions are the enhancer-promoter interactions.

**TAD fusion prediction problem:** Given a Hi-C contact frequency matrix of a chromosome and the coordinates of a deletion, we predict if this deletion results in a TAD fusion and assign a TAD fusion score to that deletion. We define a TAD fusion based on new genomic interactions created between different bins as a result of the deletion. By that, the *TAD fusion score* is defined as the expected number of additional genomic interactions created as a result of the deletion.

We propose a two-step framework for calculating the TAD fusion score of a homozygous deletion: (i) predicting a new Hi-C contact matrix $G$ of the mutated chromosome (i.e., with the deletion) given the Hi-C contact matrix $H$ of the genome without the deletion and the deletion coordinates as the inputs and (ii) comparing this predicted/new Hi-C contact matrix $G$ with the modeled Hi-C contact matrix (estimated from $H$ in step (i)) to estimate the number of new interactions created as a result of that deletion. In the next two subsections, we introduce the algorithms we have developed as part of this framework to solve each of these two subproblems.

### Predicting contact frequency matrix resulting from a deletion
In this subsection, we provide a combinatorial algorithm for predicting the contact frequency matrix $G$ of a genome

with a homozygous deletion $\Delta_{x,y}$ given the Hi-C contact frequency matrix $H$ of the genome without the deletion as the input. We assume that the contact frequency between any two bins $i, j$ will not be changed as a result of the deletion if both bins $i$ and $j$ are in the upstream or the downstream of the deletion (i.e., $G_{i,j} = H_{i,j}$ if $1 \leq i, j < x$ or $y < i, j \leq n$). Our goal is to predict the contact frequency $G_{i,j}$ between any bin $i$ in the upstream of the deletion and any bin $j$ in the downstream of the deletion (i.e., $1 \leq i < x$ and $y < j \leq n$).

### Modeling the contact frequency
For any two bins $i$ and $j$, let $H'_{i,j}$ be their contact frequency that we estimate from our model. First, it has been shown that power-law scaling based on the genomic distance best captures the contact frequencies observed in Hi-C experiments [8, 66, 70]. We denote the power-law scaling factor by parameter $\beta$ and model the Hi-C contact map as $H'_{i,j} \propto |i - j|^{\beta}$ [8, 70]. Second, it is shown that contact frequency between two bins is also affected by the genomic properties of these two bins (e.g., GC content, mappability). Therefore, in predicting the Hi-C contact frequencies using the genomic distance, we also introduce a parameter $\alpha_i$ (in the log-scale) for each bin $i$ to capture the genomic properties/biases of this bin. Thus, the model can be extended to $H'_{i,j} \propto e^{(\alpha_i + \alpha_j)/2} |i - j|^{\beta}$ [70, 71]. Finally, it is shown that the contact frequency between two bins $i$ and $j$ drops rapidly if there is a TAD boundary/insulator (e.g., a binding site of CTCF proteins) [2, 12] between these bins. Therefore, we introduce a variable $r_k$ for each bin $k$ to represent potential existence of a separator/insulator at that bin. The reduction of contacts between any two bins due to a separator/insulator at bin $k$ is modeled by an exponential functions of $r_k$ as in [72]. Thus, the Hi-C contact frequency is finally modeled as follows

$$H'_{i,j} = \frac{e^{(\alpha_i + \alpha_j)/2} |i - j|^{\beta}}{e^{r_{i+1}} e^{r_{i+2}} \ldots e^{r_j}} \qquad (1)$$

### Parameter optimization
Note that all variables $(\alpha_i, \beta, r_i)$ will be estimated by utilizing solely the input Hi-C matrix $H$. For that, we need to minimize the difference between $H_{i,j}$ and $H'_{i,j}$. However, as the contact frequencies between different bins can be orders of magnitude, instead of minimizing the absolute differences, we try to make their ratio as close to 1 as possible (i.e., $H_{i,j}/H'_{i,j} \to 1$). We can achieve this by minimizing the absolute log value of that ratio as shown below:

$$\left| \log \left( \frac{H_{i,j}}{H'_{i,j}} \right) \right| = \left| \log(H_{i,j}) - \log \left( \frac{e^{(\alpha_i + \alpha_j)/2} |i - j|^{\beta}}{e^{r_{i+1}} e^{r_{i+2}} \ldots e^{r_j}} \right) \right|$$

$$= \left| \log(H_{i,j}) - \frac{\alpha_i + \alpha_j}{2} - \beta \log |i - j| + \sum_{i < k \leq j} r_k \right| \qquad (2)$$

Thus, we can estimate all parameters by solving the optimization problem below:

$$\underset{\alpha,\beta,r}{\text{Minimize}} \quad \sum_{1 \le i \ne j \le n} \left| \log(H_{i,j}) - \frac{\alpha_i + \alpha_j}{2} - \beta \log|i-j| + \sum_{i < k \le j} r_k \right|$$

such that

$$r_k \ge 0$$
$$\beta \le 0$$

(3)

**Solving the optimization problem** We introduce slack variables $z_{ij}$ to turn the optimization problem 3 into a linear programming (LP) problem as shown below:

$$\underset{\alpha,\beta,r}{\text{Minimize}} \quad \sum_{1 \le i \ne j \le n} z_{ij}$$

such that

$$z_{ij} \ge + \left( \log(H_{i,j}) - \frac{\alpha_i+\alpha_j}{2} - \beta \log|i-j| + \sum_{i<k \le j} r_k \right)$$
$$z_{ij} \ge - \left( \log(H_{i,j}) - \frac{\alpha_i+\alpha_j}{2} - \beta \log|i-j| + \sum_{i<k \le j} r_k \right)$$
$$r_k \ge 0$$
$$z_{ij} \ge 0$$
$$\beta \le 0$$

(4)

### *Prediction of the contact frequencies*
We assume that the deletion does not change any parameter value of $\alpha, \beta, r$; it only reduces the genomic distance and remove some insulators. Thus, the contact frequency will be estimated explicitly as:

$$G_{i,j} = \frac{e^{(\alpha_i+\alpha_j)/2} |i-j-(y-x+1)|^{\beta}}{e^{r_{i+1}} \dots e^{r_{x-1}} e^{r_{y+1}} \dots e^{r_j}}$$

(5)

### Estimating the TAD fusion score of a deletion mutation
We denote the TAD fusion score of a deletion $\Delta_{x,y}$ as $S_{\Delta_{x,y}}$ and define it as the difference between the expected number of genomic interactions between bins in the genome with and without the deletion.

$$S_{\Delta x,y} = E(C_G) - E(C_H)$$
$$= \sum_{i \in [1,x-1]} \sum_{j \in [y+1,n]} w_{ij}(p(c_{i,j}=1|G) - p(c_{i,j}=1|H'))$$

(6)

$$= \sum_{i \in [1,x-1]} \sum_{j \in [y+1,n]} w_{ij} \left( p\left(c_{i,j}=1 \left| \frac{G_{i,j}}{e^{(\alpha_i+\alpha_j)/2}} \right. \right) \right.$$
$$\left. -p\left(c_{i,j}=1 \left| \frac{H'_{i,j}}{e^{(\alpha_i+\alpha_j)/2}} \right. \right) \right)$$

(7)

$$= \sum_{i \in [1,x-1]} \sum_{j \in [y+1,n]} w_{ij} \left( f\left( \frac{G_{i,j}}{e^{(\alpha_i+\alpha_j)/2}} \right) - f\left( \frac{H'_{i,j}}{e^{(\alpha_i+\alpha_j)/2}} \right) \right)$$

(8)

where random variables $C_G$ and $C_H$ represent the number of interactions between different bins assuming the Hi-C

contact matrices for the two genomes $H$ and $G$ respectively. The random variable $c_{i,j}$ is an indicator variable representing existence of the interaction between bins $i$ and $j$ (i.e., $c_{i,j} = 1$ indicates an interaction between bins $i$ and $j$ and $c_{i,j} = 0$ indicates the lack of such interaction in 3D space). We also added the weight $w_{ij}$ for important interactions (e.g., between enhancers and promoters), but in all analyses, we simply set $w_{ij} = 1$. Here, we use the normalized value (i.e., $G_{i,j}/e^{(\alpha_i+\alpha_j)/2}$) rather than the raw value $G_{i,j}$ to eliminate the bias so that we can compare the TAD fusion score between different datasets. We assume that the genomic interaction probability of a pair of bins only depends on their normalized contact frequency. Therefore, we can use a function $f$ to convert this normalized contact frequency to the genomic interaction probability for both $G$ and $H'$. Here we simply set $f$ as a step function with a threshold $\delta$ (i.e., $f(x) = 1$ if $x \ge \delta$, otherwise $f(x) = 0$). To determine this threshold value, we used the HiCCUP loop set [42]. We assumed that two anchors of a loop have a genomic interaction. Then, we calculated the normalized contact frequency of these interacted locus pair and set $\delta$ as the minimum value. By that, we assumed that any pair of loci in which the normalized contact frequency is greater or equal to that minimum value will interact; otherwise, this pair is considered non-interact. Notice that, in above formula (Eqs. 6, 7, and 8), we use the fitted value $H'_{i,j}$ since the raw value $H_{i,j}$ may contain noise. By that, we can guarantee that the score $S_{\Delta_{x,y}}$ is always positive since $G_{i,j}$ is always less or equal to $H'_{i,j}$ (see Eqs. 1 and 5).

### Implementation
The linear programming problem (Eq. 4) has $O(n^2)$ variables where $n$ is the number of bins of a chromosome. Assuming for one human chromosome we have over 40,000 bins (e.g., chromosome 1 with bin length 5 kbp), this results in over $10^9$ variables. This large size problem is not efficiently solvable with our current LP solvers (e.g., CPLEX). Thus, we employed a strategy to reduce the running time in practice by making some compromises as follows: (i) in regard to estimating the parameters, we partitioned each chromosome into smaller overlapping segments (600 bin segments with 50 bin overlaps) and estimated the parameters for each segment; (ii) in the objective function of the LP problem (Eq. 4), we only considered the summation of bin pairs that the genomic distance was at most 50 bins apart (i.e., 2Mbp for Hi-C dataset [49], to evaluate the modeled Hi-C data with long-range interactions), 10 bins apart (i.e., 400 kbp for Hi-C dataset [49], to analyze the insulation at each bin), and 100 bins apart (i.e., 500 kbp for GM12878 Hi-C data [42], to analyze the deletions); (iii) the parameter $\beta$ is limited to be in the range of the reported values in the literature ($-2 \le \beta \le -1$) [8, 73]; and (iv) for estimating

the TAD fusion score of deletions (great ape lineages, 1KG, TCGA, diseases), we only consider the interaction change between the 500 kbp region upstream and the 500 kbp downstream of the deletion. We have performed an extensive simulation to evaluate the robustness of our method under different ranges of input parameters (see Additional file 1: Figure S7). Note that although larger values of these parameters provide a slightly more accurate result, these parameter settings make the running time of the method increase significantly. Thus, we are compromising the accuracy and efficiency in our parameter value selection. Our program was run in 48 h (on a cluster with 32 CPUs and 80 GB memory) to fit the model for all 23 chromosomes of GM12878, which was done only once. After all parameter values were estimated, calculating the TAD fusion score for any deletion was less than 1 s per one deletion.

## Additional files

**Additional file 1:** Supplementary figures and tables. (PDF 10,614 kb)

**Additional file 2:** Review history. (PDF 2003 kb)

### Abbreviations
1KG: 1000 genomes project; 3C: Chromosome conformation capture; 3D: Three-dimensional; CRISPR: Clustered regularly interspaced short palindromic repeats; CTCF: CCCTC-binding factor; LP: Linear programming; SCC: Stratum-adjusted correlation coefficient; SV: Structural variation; TAD: Topologically associating domains; TCGA: The Cancer Genome Atlas

### Acknowledgements
We would like to thank Ali Aliyari for his help with the first screening result and Farhad Hormozdiari for his useful comments. We would like to thank the reviewers for their constructive comments (Additional file 2).

### Funding
This work is supported in part by the Sloan Research Fellowship number G-2017-9159 to FH.

### Availability of data and materials
TAD-fusion score is written in C++. The source code is freely available on GitHub at https://github.com/HormozdiariLab/TAD-fusion-score [74] under the BSD-2 license. The version used in the manuscript is deposited in zenodo https://doi.org/10.5281/zenodo.2574383 [75]. Other datasets and tools used in this study are available at the links listed in the supplemental information (Additional file 1: Table S3).

### Authors' contributions
LH and FH developed the method, analyzed the data, and wrote the manuscript. LH implemented the TAD fusion score software. Both authors read and approved the final manuscript.

### Ethics approval and consent to participate
Not applicable.

### Consent for publication
Not applicable.

### Competing interests
The authors declare that they have no competing interests.

## 

### Author details
[1] Genome Center, UC Davis, Davis, USA. [2] UC Davis MIND Institute, Sacramento, USA. [3] Department of Biochemistry and Molecular Medicine, UC Davis, Sacramento, USA.

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
