## [Review history. (PDF 2003 kb) · Genome Biology]

First round of review

Reviewer 1

Are the methods appropriate to the aims of the study, are they well described, and are necessary controls included?

Yes

Are the conclusions adequately supported by the data shown?

Yes

Are sufficient details provided to allow replication and comparison with related analyses that may have been performed?

Yes

Does the method perform better than existing methods (as demonstrated by direct comparison with available methods)?

No: Not applicable

Is the method likely to be of broad utility? Is any software component easy to install and use?

Haven't tested the software but the results and framework is likely of broad utility.

Is the paper of broad interest to others in the field, or of outstanding interest to a broad audience of biologists?

Yes: It is an important line of work to systematically predict the impact of deletions on genome architecture.

Comments to authors

This manuscript describes a method for scoring deletions based on their potential impact on the 3D structure and TAD boundaries. Methodologically, it provides a nice formulation to this problem, which is novel aspect. The experimental results also look quite convincing that the devised score ranks disease-related deletions higher than those in 1KG project. However, there are some important points to be addressed before this manuscript is appropriate for publication as I detail below.

1. Li et al. in their 2016 Scientific Reports paper (3Disease Browser) talk about an algorithm to predict disease-associated chromosomal rearrangement events that effect TADs. They describe Structure Influence score to measure the impact of structural variants using Hi-C data (in multiple cell types) as well as ChIP-seq data to characterize enhancers. This work should certainly be cited and compared against when appropriate.

2. I am not clear on the fact that how the authors are distinguishing a TAD-fusion causing deletion from a deletion that does not create a fusion but involves a boundary region while calculating the score. Franke et al. in their 2016 paper (extended fig. 8) showed a boundary deletion occurring between Kcnj2-TAD and Sox9-TAD, but the boundary deletion caused no change in overall TAD structure while it did change the expression of Kcnj2 gene.

3. Regarding fig. 2a: This brings the question that whether a simple score aggregating results of TAD prediction methods would equally well rank the importance of deletions compared to TAD fusion. Similar question can be asked for a single score (e.g., insulation score) that measures insulating capacity of a

given region.

4. Fig. 4d: The figure shows a strong positive correlation with the length of the deletion event. It is evident that a higher length deletion event has more potential to cause 3D structural damage than a smaller one especially when the impact is measured by the score proposed here by the authors. Could the authors clarify how much of the nice experimental results they have (such as Fig 4a) is beyond this simple correlation with length?

5. While the authors describe their score as "the expected total number of changes in pairwise genomic interactions", it would be important to treat interactions that involve regulatory (promoters, enhancer) or insulation regions differently than other regions.

6. The architecture of 3D genome varies according to cell type even though TADs are conserved to a certain extent. It is not unexpected to see more severe consequences of a deletion in one cell type compared to another (a development related enhancer or a gene deleted in stem cells versus terminally differentiated cells). It may be important to at least discuss this or address it by using Hi-C data from several cell types.

7. The authors may want to consider better models to identify interacting pairs. The assumption stated in "We assume that any two bins i and j that are inside the same loop will have a genomic interaction and if they belong to two separate loops then they are not interacting." is certainly problematic and not in line with the literature.

Reviewer 2

Are the methods appropriate to the aims of the study, are they well described, and are necessary controls included?

See comments below.

Are the conclusions adequately supported by the data shown?

See comments below.

Are sufficient details provided to allow replication and comparison with related analyses that may have been performed?

Yes

Does the method perform better than existing methods (as demonstrated by direct comparison with available methods)?

No: See comments below.

Is the method likely to be of broad utility? Is any software component easy to install and use?

Not sure, see comments below

Is the paper of broad interest to others in the field, or of outstanding interest to a broad audience of biologists?

Yes: If proven to be better than other approaches, it will be very useful to the 3D community.

Comments to authors

In this study, the authors describe a method to computationally score TAD fusion events due to deletions. While the approach is interesting, there are several aspects of this study that would need significant improvements to make it suitable for publication.

1. Computational validation of proposed method

The proposed method first models the Hi-C matrix based on distance between bin pairs, genomic features of each bin and presence of insulators. After this step, the Hi-C matrix model is utilized to assess the expected changes in connectivity upon deletion events. This approach is interesting but needs to be extensively validated. More specifically, how good is the approximation of the Hi-C matrix? Do the predicted locations of insulation match experimental data, i.e. CTCF ChIP-seq sites in the same cell type/line? A systematic evaluation needs to be performed across multiple Hi-C/CTCF datasets:

- a. Compare modeled Hi-C matrix to the actual matrix in multiple Hi-C datasets using distance-adjusted correlations such as HiCRep.
- b. A subtle point here is how do we know that any correlation value is good enough. Therefore, the authors will need to think of the right controls. For example, I would propose picking two closely related cell types and comparing actual and estimated Hi-C matrices in all pair-wise combinations, again using a method like HiCRep. The estimated Hi-C matrix should more closely approximate the actual Hi-C matrix from the same cell type. An ideal dataset for this analysis would be the stem cell dataset from Bing Ren's lab: <https://www.nature.com/articles/nature14222>
- c. Compare predicted insulation loci to actual CTCF data, again across multiple datasets. Again, perform the analysis described in (b) to make sure that the approximation is good enough to capture subtle changes from cell type to cell type.

2. Comparison of proposed method to other approaches

Previous work (<https://www.ncbi.nlm.nih.gov/pubmed/28056762>) has shown that some TAD callers (e.g. Armatus) do not generate reproducible TAD boundaries. Therefore, including such inconsistent TAD callers in the consensus TAD boundary calling is not appropriate. Also, what was the bin size used for TAD calling? Using a 5kb resolution may introduce unnecessary noise, and I suggest repeating the analysis at a "standard" 40kb resolution. More importantly, precisely because of issues with TAD callers in general, as pointed out by the authors and several studies, it is problematic to use them as a comparison to the proposed method. Instead, the authors can use the concept of insulation score, as computed for each bin (not just at TAD boundaries). Then, their evaluation will not be dependent on whether the TAD boundary was called or not. The hypothesis here is that the higher the maximum insulation score inside the deletion region, the higher the probability of fusion. Therefore, this score should correlate with the proposed TAD fusion score.

3. Length of deletions

Some deletions in cancer can be really big, megabases long. Have the authors applied any upper limit? Figure 4d suggests so, but I am not entire sure if this is explicitly mentioned in the paper. Also, in Figure 4d, there is a puzzling association of TAD fusion score with the size of the deletion. The authors need to explain why this is real, or, alternatively, look into their scoring approach for ways to normalize for the size. For example, I am not sure whether normalization over the size of the "rectangle" used for scoring

has been done. After, this normalization, Figure 4c will probably be affected and this may change their conclusions.

Other comments:

- a. Lines 26-27: These contacts are still between bin pairs of variable distance, so HiCRep should be used here as well instead of regular Pearson correlation.
- b. Line 34: Armatus is misspelled
- c. Figure S1, clearly mark the rectangle of where that TAD fusion score is calculated on the Hi-C heatmaps

Introduction:

- a. Line 44-46: Reference [24], in agreement with Lazaris et al, BMC Genomics 2017, shows that some TAD callers are quite consistent, whereas calling specific loops is a much harder problem. Therefore, the authors should appropriately rephrase.
- b. Line 47-48: Other reasons for lack of high concordance can be sequencing depth, and differences in library preparation (e.g. different cutting enzymes)
- c. Line 59: the disruption in the PDGFRA locus is due to methylation, not deletion (reference [42]).
- d. Two more papers to cite: [https://www.cell.com/cell/fulltext/S0092-8674\(18\)31021-3](https://www.cell.com/cell/fulltext/S0092-8674(18)31021-3) and <https://www.ncbi.nlm.nih.gov/pubmed/27869826>
- e. Negative selection of deletions of TAD boundaries in a pan-cancer analysis has been recently demonstrated in <https://www.ncbi.nlm.nih.gov/pubmed/29416042>. Actually, this paper may be useful for computing the insulation scores (proposed in my comments above), as it shows a strong association between insulation score, CTCF binding strength and negative selection of deletions.

Reviewer #1

This manuscript describes a method for scoring deletions based on their potential impact on the 3D structure and TAD boundaries. Methodologically, it provides a nice formulation to this problem, which is novel aspect. The experimental results also look quite convincing that the devised score ranks disease-related deletions higher than those in 1KG project. However, there are some important points to be addressed before this manuscript is appropriate for publication as I detail below.

Response: We thank the reviewer for their positive comments. We have done extensive experiments, added more evaluation data and revised the manuscript as the reviewer suggested.

1. Li et al. in their 2016 Scientific Reports paper (3Disease Browser) talk about an algorithm to predict disease-associated chromosomal rearrangement events that effect TADs. They describe Structure Influence score to measure the impact of structural variants using Hi-C data (in multiple cell types) as well as ChIP-seq data to characterize enhancers. This work should certainly be cited and compared against when appropriate.

Response: We thank the reviewer for pointing us this relevant work. Although 3Disease Browser also predicts if a structural variant can affect the TAD structure, it has two limitations:

(i) It estimated the TAD fusion probability by using the max insulation score of deleted bins. We show in the comment 3 below that TADs are usually overlapped and the boundary may have more than one strong insulator in several adjacent bins. Therefore, although max insulator score is large, it does not guarantee the deletion removes all strong insulators at the boundary.

(ii) More importantly, this method can not predict how region/loci interactions will be changed (as we did in the case of Kcnj2-Sox9 boundary deletion in the comment 2 below)

Furthermore it seems the web server only has a set of predefined deletion which can be queried, but no new deletions can be test on the web server and the method itself seems to not be publicly available to run. We have cited this paper in the introduction as:

“For example, Li et al. introduced the Structure Influence score but their method did not show specifically how chromatin interaction might be changed due to a given deletion.”

2. I am not clear on the fact that how the authors are distinguishing a TAD-fusion causing deletion from a deletion that does not create a fusion but involves a boundary region while calculating the score. Franke et al. in their 2016 paper (extended fig. 8) showed a boundary deletion occurring between Kcnj2-TAD and Sox9-TAD, but the boundary deletion caused no change in overall TAD structure while it did change the expression of Kcnj2 gene.

Response: Again we thank the reviewer for pointing out this nice example. One of the motivations behind developing our method is to distinguish the real TAD fusions from deletions

which remove TAD boundary but do not cause a significant TAD fusion. We argue that an intuitive way to solve this problem is to first predict how a deletion alters the the 3D structure of genome and then quantify if the alteration in the 3D structure is indicative of two seperate TADs being fused. Thus, our proposed approach in theory should be able to give lower scores to deletions which remove a TAD boundary but don't cause TAD fusion versus the ones that cause TAD fusion.

In this special example, we used our model to predict the change of chromatin interaction due to the deletion. Our Hi-C matrix prediction totally agrees with the real experimental Hi-C data produced after the deletion reported by Franke et al. (HiCRep score between our predicted Hi-C matrix and the real Hi-C after deletion was 0.94). In figure below the first panel (WT) is the real Hi-C matrix before the deletion, the second panel (Del) is the real data Hi-C matrix after deletion, the third panel (Prediction) is the Hi-C matrix predicted using our method and the final panel (Prediction - Del) indicates the difference between our prediction and the real data. As noted and can be seen in the figure below our prediction of Hi-C matrix is in complete agreement with what was observed using real data after deletion (HiCRep score of 0.94).

We observed that there was a very small TAD (or sub-TAD, on the downstream of the deletion) that was fused to a large TAD on the upstream of the deletion. This is in an agreement with what can be seen from real Hi-C data matrix (the second panel (Del)). This small fusion generated new interactions (marked by dashed lines in first three sub-figures) that might have caused the increase of Kcnj2 expression. We added this case study in the manuscript (Supplementary Figure S4, lines 139 - 149). The modification to the main manuscript and the new figure (Supplementary Figure S4) is as follow:

“Finally, we investigated the performance of our proposed approach for predicting Hi-C contact frequencies after deletions against the experimental Hi-C data for the same set of deletions. We used deletions at Sox9-Kcnj2, HoxD and Firre regions where Hi-C data existed for this comparison (Franke et al. Nature 2016, Rodriguez et al. Gene & Dev. 2017, Barutcu et al. Nat Comm 2018). We compared the predicted Hi-C data and the experimentally measured Hi-C data (see Supplementary Figures S3, S4). Since these interactions were at variable distance, again we used the stratum adjusted correlation coefficient (SCC) from HiCRep package (Yang et al. GR 2017) to evaluate the similarity instead of regular Pearson correlation coefficient. For the three deletions (at Sox9-Kcnj2, HoxD and Firre), our prediction Hi-C data was quite similar to the experimental Hi-C data of these deletions with an average SCC of 0.82. Interestingly, in the case of the deletion at the Sox9-Kcnj2 boundary, we observed that there was a very small TAD (or sub-TAD, on the downstream of the deletion) that was fused to a large TAD on the upstream of the deletion (as can be seen using both our predicted Hi-C matrix and the real experimental Hi-C matrix in Supplementary Figure S4). This fusion might cause the increase of Kcnj2 expression reported in (Franke et al. Nature 2016).”

3. Regarding fig. 2a: This brings the question that whether a simple score aggregating results of TAD prediction methods would equally well rank the importance of deletions compared to TAD fusion. Similar question can be asked for a single score (e.g., insulation score) that measures insulating capacity of a given region.

Response: We can not confirm this conclusion since the aggregated score will be resulted of applying some heuristic function on an intersection of different heuristics (i.e., TAD callers). Furthermore, to best of our knowledge there is no tool which combines the union of several TAD callers into a score for TAD fusion. Furthermore, in theory the TAD fusion is more than just deletion of TAD insulators/boundaries (as shown in figure below from a 1KG example). Currently, our Figure 2a (now has become supplemental figure S5 in the paper) simply confirms two major points: (i) the TAD structure/boundaries predicted can significantly vary between different TAD callers as it has been reported by previous studies, and (ii) this result also provides a simple verification that our tool gives a large score to deletions which are very clearly a TAD fusion, i.e. all (or most) TAD caller tools agree the deletion is indeed deleting a TAD boundary. We corrected the manuscript to clarify this confusion (lines 152 - 161). The modification is as follow:

“The first comparison was against different methods for TAD prediction (Rao et al. Cell 2014, Zhan et al. GR 2017, Crane et al. Nature 2015). We observed that the TAD boundaries predicted varied significantly between different TAD callers and there was only a small fraction of consensus boundary predicted between all the tools (Supplementary Figure S5). Interestingly, we observed that the average TAD fusion score calculated by our approach was higher for deletions which were predicted to remove a TAD boundary based on majority of the TAD callers tested (Supplementary Figure S5).”

A single score (e.g. the sum/max of all insulation score of all bins of a deletion as 3Disease Browser did) may not be correct to predict the TAD fusion in some cases (as shown in example below from 1KG). First reason is that TADs are usually overlapped and the *boundary may have more than one peak* in several adjacent bins (as in the figure below). Therefore, although the sum/max value is large, *it does not guarantee the deletion removes all peaks at the boundary*. Alternatively, we can use a similar approach as our approach by checking each loci pair to see if the deletion removes all insulators. But in this case, the prediction is still heuristic since we do not have a consistent model for both estimating the insulation score and predicting the interaction from that insulation score. Second, we are quantifying the TAD fusion which is not only dependent on the strength of the boundary being deleted but also the length and strength of the TADs being fused (i.e., in our model fusing two large TADs will be given a higher score than fusing two small TADs). Third, our model and method not only provide a score for quantifying the TAD fusion, but also provide the prediction of how the 3D structure will change as a result of the deletion. We clarified this point (lines 162 - 171, Supplementary figure S6) in the manuscript as follow:

“The second approach we utilized for comparison was to use the maximum LRI score (i.e., an insulation score predicted by (Chen et al. NAR 2018) of the deleted bins to rank/score each deletion. We did observe a significant yet moderate correlation between the TAD fusion score assigned to each deletion and the maximum LRI score assigned to any bin inside each deletion ($r = 0.43$, $p < 1e-10$, for chromosome 1). Further investigation has revealed that some of the discordance between calculated TAD fusion score and the maximum insulation score (LRI) inside the deletions came from the cases where there were more than one strong insulator near the TAD boundary. In these cases, some of insulators were not deleted/alterd as a result of the deletion (see Supplementary Figure S6). Thus, the deletion of a bin with high insulation score might not result in TAD fusion in these cases. Our method avoided this limitation by taking into account both the deleted insulators and the remaining (i.e. not deleted) insulators rather than considering only deleted ones.”

An illustrated example (one of 1KG deletions at chromosome 1) shows the limitation of the scoring method that only uses the maximum insulation score of bins inside the deletion. In that case, this method gives a high score since the deletion removes a bin with a large insulation score (measured by LRI score or our insulation). Our method gives a low score for this deletion since there are still other insulators remaining (both the upstream and downstream of the deletion). The upper image is visualized with Juicebox, the lower image visualizes the LRI score and our insulation that are zoomed in around the deletion.

4. Fig. 4d: The figure shows a strong positive correlation with the length of the deletion event. It is evident that a higher length deletion event has more potential to cause 3D structural damage than a smaller one especially when the impact is measured by the score proposed here by the authors. Could the authors clarify how much of the nice experimental results they have (such as Fig 4a) is beyond this simple correlation with length?

Response: We thank the reviewer for this comment and providing an opportunity to clarify the correlation between deletion length and TAD fusion score. Indeed there is a correlation between TAD fusion score and the length of the deletion, which is very much expected. First, longer deletions will result in two loci which were much farther apart to get adjacent and cause modification in 3D structure and more importantly a longer deletion has a higher chance of deleting a TAD boundary. However, we would like to point out that all of the biological results reported in the paper has been replicated after taking significant steps to remove the length bias from our analysis (results reported in Figure 4b and 4c are based on binning the deletions based on their length).

Although considering the significant difference in length of disease deletions versus 1KG set in Figure 4a, we can not compare these diseased deletions with 1KG deletions at the same length range to see the comparison result after removing the length bias. However, it is possible to infer some shorter deletions (from the original diseased deletions) that probably cause the disease as reported here (and Figure 4b). For example, Lupianez et al. (Cell 2015, PMID:25959774) reported a deletion DelB (chr2: 221278232-223014332, hg19) that caused the limb malformation disease. This deletion was 1.3Mb long and they also confirmed an equivalent deletion in mice (chr1:76388978-78060839, mm9) that caused this disease. However, they also validated a control deletion at chr1:76388978-77858974 and they confirmed that “animals carrying these deletions had normal limbs and did not show any other abnormalities”. Therefore, we inferred that the deletion chr1:77858974-78060839 might cause the limb malformation (the remaining part of the disease deletion after excluding the deletion which does not cause the disease). We lifted over this mm9 coordinates to get the hg19 coordinates of chr2:222800125-223011646. This inferred deletion was only 212kb and it can be compared with some 1KG deletions at the same length range (See Figure below and the Figure 4b in manuscript).

We inferred four such deletions (see Table S5). We compared these deletions with 1KG deletions at the same length range. We observed a significant increase of the TAD fusion score of diseased deletions against the score of 1KG deletions (see figure below). We also merged

figure 4c and 4d (now became 4c) to make a comparison between deletions of GreatApe, 1KG and TCGA at the same length range. As it can be seen in the figure 4c, even considering the deletions of the same length the TAD fusion score is highest for TCGA, followed by 1KG and then GreatApe. We updated our manuscript (lines 200 - 233, figure 4b and figure 4c) to address the deletion length issue.

5. While the authors describe their score as "the expected total number of changes in pairwise genomic interactions", it would be important to treat interactions that involve regulatory (promoters, enhancer) or insulation regions differently than other regions.

Response: We completely agree with the reviewer that adding a higher weight to interaction of regulatory elements is a very nice feature. Furthermore, our method and formulation can trivially be extended to allow such an extension (weighted expectation in equation 6 instead of expectation). For example, using such a feature we can rank the deletion better (e.g. the Sox9-Kcnj2 deletion that the reviewer suggested above). We tried this extension, however, we realized that the results can be very dependent on the enhancers being considered and can have different results from using different approaches and tools for enhancer predictions as in the recent benchmark of ML. Capra et al 2018 ("Genome-wide Enhancer Maps Differ Significantly in Genomic Distribution, Evolution, and Function", <https://www.biorxiv.org/content/early/2017/08/15/176610>). Furthermore, we are hoping that the current results reported to be independent from any additional data (such as enhancer

prediction) as input. Therefore, we corrected our formula to add the weight for each interaction as desired by the user. However, we still kept the old analysis (i.e. set all weights to 1) to avoid the bias by using incorrect weights. We leave this important extension as a future work until we have better data/approaches for characterizing enhancers. We corrected the manuscript to clarify this extension (lines 364-365 and equation 6 of the paper) as *We also added the weight w_{ij} to implementation to indicate the importance of interactions (e.g. between enhancer and promoter) in the formulation. However, in the current analysis to make the result not dependent to any additional input (e.g., predicted enhancer) we have simply set all $w_{ij} = 1$.*

6. The architecture of 3D genome varies according to cell type even though TADs are conserved to a certain extent. It is not unexpected to see more severe consequences of a deletion in one cell type compared to another (a development related enhancer or a gene deleted in stem cells versus terminally differentiated cells). It may be important to at least discuss this or address it by using Hi-C data from several cell types.

Response: We thank the reviewer for pointing out this suggestion. We agree with the reviewer that the deletions of interest should be tested using different cell type Hi-C data to make sure we can find and rank deletions which might be causing TAD-fusion in on cell type and not others. We have added the following paragraph to the discussion section of the paper (line 267-273) to address this point:

“It is important to note that although the TADs are believed to be conserved in most cell types, however there still might be important difference in genome structure between different cell types or different stages of development. This difference should be reflected in the Hi-C matrix of that cell type. Furthermore, while one deletion might not cause a significant genome structure disruption in one cell type it might do so in another cell type. Thus, as more high quality Hi-C data of different cell types are produced, the deletions should be tested and scored using each of the Hi-C matrices. By that we can maximize the possibility of finding deletions which might cause TAD fusion in one cell type or a specific development stage.”

7. The authors may want to consider better models to identify interacting pairs. The assumption stated in "We assume that any two bins i and j that are inside the same loop will have a genomic interaction and if they belong to two separate loops then they are not interacting." is certainly problematic and not in line with the literature.

Response: Our assumption was simply from the concept of TADs where two bins belong to the same TAD will interact with each other while two bins belong to two different TADs will not interact. As the reviewer suggested, we simplified our model where we assumed that only two anchors of a loop interacted (determined and validated in Rao 2014 's paper). We calculated the normalized contact frequency of these interacted locus pair and took the minimum value. Then we assumed that any pair of loci that the normalized contact frequency is greater or equal to that minimum value will interact, otherwise this pair is considered non-interact. With that model, we observed that the TAD fusion score was similar to the old one ($PCC > 0.7$) and all the

biological conclusions remained the same as before. We updated the results (Figures 3, 4, S3, S4, S5, S7) and clarified this simplified model in the manuscript (lines 368 - 374) as

“Therefore, we can use a function f to convert this normalized contact frequency to the genomic interaction probability for both G and H . Here we simply set f as a step function with a threshold δ (i.e. $f(x) = 1$ if $x \geq \delta$, otherwise $f(x) = 0$). To determine this threshold value, we used the HiCCUP loop set (Rao et al., 2014). We assumed that two anchors of a loop have a genomic interaction. Then we calculated the normalized contact frequency of these interacted locus pair and set δ as the minimum value. By that, we assumed that any pair of loci that the normalized contact frequency is greater or equal to that minimum value will interact, otherwise this pair is considered non-interact.

Reviewer #2

In this study, the authors describe a method to computationally score TAD fusion events due to deletions. While the approach is interesting, there are several aspects of this study that would need significant improvements to make it suitable for publication.

Response: We thank the reviewer for his thoughtful comments to improve the paper. We added more experiments and evaluations to the paper as suggested by the reviewer.

1. Computational validation of proposed method

The proposed method first models the Hi-C matrix based on distance between bin pairs, genomic features of each bin and presence of insulators. After this step, the Hi-C matrix model is utilized to assess the expected changes in connectivity upon deletion events. This approach is interesting but needs to be extensively validated. More specifically, how good is the approximation of the Hi-C matrix? Do the predicted locations of insulation match experimental data, i.e. CTCF ChIP-seq sites in the same cell type/line? A systematic evaluation needs to be performed across multiple Hi-C/CTCF datasets:

a. Compare modeled Hi-C matrix to the actual matrix in multiple Hi-C datasets using distance-adjusted correlations such as HiCRep.

Response: As suggested by the reviewer, we used the stratum adjusted correlation coefficient (SCC) of the HiCRep package to evaluate our approximation of Hi-C matrices. We used the Hi-C datasets from Bing Ren's lab (Schmitt et al, Cell reports 2016). We compared raw Hi-C matrices and our predicted Hi-C matrices (using our proposed approximation model) on chromosome 1 (the longest chromosome). As we expected, we observed that the SCC score between a raw Hi-C matrix (denoted as *raw*) and our modeled matrix (denoted as *model*) was very high for all the seven matching *raw* versus *model* pairs (ranging from 0.948 to 0.976, as reported in table below). More importantly, our modeled Hi-C matrix for each cell type was closest (highest SSC score) to the raw Hi-C matrix of the same cell type (the bold values in the diagonal of the table below). Conversely, the raw Hi-C matrix was also the closest one to the estimated Hi-C matrix of the same cell type. This experiment as requested by the reviewer indicates that our method is capable of accurately capturing the subtle difference between cell types from different Hi-C matrices. We added this evaluation in the manuscript (lines 104 - 113, supplementary table S4). The manuscript was modified as follows:

"We first evaluated the capability of our proposed model in producing the approximate Hi-C data matrix using the Hi-C datasets (of chromosome 1) of seven different cell lines from (Schmitt et al. Cell Reports 2016) as input data (see Supplemental Table S4). We compared raw Hi-C matrices (denoted as {it Raw}) and the Hi-C matrices produced using our proposed method (denoted as {it Model}). The stratum adjusted correlation coefficient (SCC) of the HiCRep package (Yang et al. GR 2017) was used to estimate the similarity between any pair of Hi-C matrices (i.e. {it Raw} versus {it Model}). We observed that the SCC score between a raw Hi-C

matrix and our modeled matrix was very high for all the seven matching raw versus model pairs (ranging from 0.948 to 0.976, see Supplemental Table S4).

More importantly, our modeled Hi-C matrix of each cell line was closest (highest SSC score) to the raw Hi-C matrix of the same cell line (the bold values on the diagonal of the Supplemental Table S4). Conversely, the raw Hi-C matrix was also the closest one to the modeled Hi-C matrix of the same cell line. This indicated that our method was capable of accurately capturing the subtle difference between cell lines from their Hi-C data matrices.“

	H1 (model)	ME (model)	NP (model)	TP (model)	MS (model)	IMR90 (model)	GM12878 (model)
H1 (raw)	0.976	0.951	0.924	0.894	0.888	0.843	0.707
ME (raw)	0.942	0.973	0.901	0.924	0.883	0.865	0.751
NP (raw)	0.936	0.922	0.972	0.878	0.882	0.839	0.711
TP (raw)	0.881	0.922	0.85	0.973	0.88	0.881	0.813
MS (raw)	0.873	0.882	0.86	0.889	0.948	0.892	0.728
IMR90 (raw)	0.821	0.853	0.809	0.879	0.879	0.949	0.761
GM12878 (raw)	0.69	0.743	0.669	0.809	0.713	0.755	0.972

b. A subtle point here is how do we know that any correlation value is good enough. Therefore, the authors will need to think of the right controls. For example, I would propose picking two closely related cell types and comparing actual and estimated Hi-C matrices in all pair-wise combinations, again using a method like HiCRep. The estimated Hi-C matrix should more closely approximate the actual Hi-C matrix from the same cell type. An ideal dataset for this analysis would be the stem cell dataset from Bing Ren's lab: <https://www.nature.com/articles/nature14222>

Response: We thank the reviewer for this helpful comment. As suggested by the reviewer, we examined further two pairs of most closely related cell types H1 vs ME and MS vs IMR90 (based on results and data from the recent paper by Bing Ren's lab (Dixon et al, Nature 2015)). In addition to the comparison between raw Hi-C matrices and modeled Hi-C matrices, we also made the comparison between raw Hi-C matrices (i.e. raw vs raw) and between modeled Hi-C matrices (i.e. modeled vs modeled) of these close cell types.

As we observed in the above figure, our estimated Hi-C matrix (our model) was closest one to the raw Hi-C matrix of the same cell type based on SCC score (HiCRep). Conversely, the raw Hi-C matrix was also the closest one to the estimated Hi-C matrix (our model) of the same cell type. This was in complete agreement with what we were expecting and was suggested by the reviewer. We also observed that our model reduced the difference between two cell types in comparison of the raw Hi-C data (i.e. two estimated matrices were closer to each other than two raw ones were). This was expected since our approximation formula only took into account the length and the insulation while other details such as the loop of some specific loci (i.e. “peak” as in Rao 2014 ‘s paper) were not utilized. We have added this new results (lines 104 - 121) to the main section of the paper and added the above figure as the one of the main subfigures in the paper (figure 2a in the paper). The manuscript was modified as follows:

“Furthermore, we also examined further two pairs of closely related cell lines H1 vs ME and MS vs IMR90 (based on results and data from (Dixon et al. Nature 2015) which includes HiC with 40kbp resolution). As depicted in Figure 2a, our estimated Hi-C matrix (model) was closest one to the raw Hi-C matrix of the same cell type based on SCC score. Conversely, the raw Hi-C matrix was also the closest one to the estimated Hi-C matrix (our model) of the same cell type. We also observed that our model reduced the difference between two cell lines in comparison of the raw Hi-C data (i.e. two estimated matrices were closer to each other than two raw ones were). This was expected since our approximation formula only took into account the length and the insulation while other details such as the loop of some specific loci (i.e. “peak” as in (Rao et al. Cell 2014)) were not utilized.”

c. Compare predicted insulation loci to actual CTCF data, again across multiple datasets. Again, perform the analysis described in (b) to make sure that the approximation is good enough to capture subtle changes from cell type to cell type.

Response: As requested by the reviewer we first compared our predicted insulation at each bin (i.e. parameter r_i) with the CTCF peak at that bin. We evaluated this for three different cell lines H1, IMR90 and GM12878 (where CTCF data was available from ENCODE). For each cell line, we grouped the bins into six groups based on the insulation (from highest to lowest scores, top 100 highest insulation, then top from 101st to 300th and so on). As expected we did observe a significant enrichment of CTCF (by the average number of peaks) at bins predicted to have

higher insulation (parameter r_i) by our proposed model (please see figure below). Furthermore, this observed enrichment holds for all three cell lines and Hi-C matrices tested. We added this CTCF enrichment validation to the manuscript (lines 122 - 129, figure 2b).

For the comment “Again, perform the analysis described in (b) to make sure that the approximation is good enough to capture subtle changes from cell type to cell type”. If we are understanding correctly, the reviewer is asking to verify if our predicted insulation can capture subtle changes from cell type to cell type. For that validation, we compared the predicted insulation at each bin (i.e. parameter r_i) for seven different cell lines from Dixon et al. (i.e., H1, ME, TB, NP, MS, IMR90 and GM12878). We built a pairwise similarity matrix based on the correlation between r_i values as shown in the figure below. The hierarchical clustering tree built from this similarity matrix was comparable to the topology built using A/B compartments reported in Dixon et al. 2015 with one minor exception of swap in the order of TB and NP. Furthermore, our clustering is identical as the one reported in Schmitt et al., Cell reports 2016. Other important features were similar to previous studies (Dixon et al. 2015, Schmitt et al. 2016, Gong et al. 2018) such as (H1, ME) and (MS, IMR90) were grouped as pairs while GM12878 was separated as an outbranch in the tree. This experiment indicated that the insulation calculated using our method was capable of capturing the subtle difference between cell types and correctly grouped more closely related cell types together (e.g., [H1, ME]). We added this validation in the manuscript (lines 129 - 138, Supplementary Figure S2). The manuscript was updated as follows:

“Third, we also compared the predicted insulation at each bin (i.e., parameter r_i) for seven different cell lines from (Dixon et al. Nature 2015) (i.e., H1, ME, TB, NP, MS, IMR90 and GM12878). We built a pairwise similarity matrix for these seven cell lines using the insulation values at each bin (Supplementary Figure S2). The hierarchical clustering tree built from this similarity matrix was comparable to the one built using A/B compartments reported in (Dixon et

al. Nature 2015) using the same set of cell lines (with one minor exception of swap in the order of TB and NP). Furthermore, the hierarchical clustering produced by the similarity matrix was in complete concordance with the clustering reported in (Schmit et al. Cell Rep. 2016). Other important features were similar to previous studies (Dixon et al. Nature 2015, Schmit et al. Cell Rep. 2016, Gong et al. Nature Comm 2018) such as (H1, ME) and (MS, IMR90) were grouped as pairs while GM12878 was separated as a outbranch in the tree. This experiment indicated that the insulation (i.e., parameter $\$r_i\$$) calculated using our method was capable in capturing the subtle difference between cell types and correctly grouped more closely related cell types together (e.g., [H1, ME]).“

2. Comparison of proposed method to other approaches

Previous work (<https://www.ncbi.nlm.nih.gov/pubmed/28056762>) has shown that some TAD callers (e.g. Armatus) do not generate reproducible TAD boundaries. Therefore, including such inconsistent TAD callers in the consensus TAD boundary calling is not appropriate.

Response: As suggested by the reviewer, we removed Armatus from the consensus analysis (the updated figure is now Supplemental Figure S5). The overall result was still the same as before where the TAD fusion prediction varied a lots by using different TAD caller tools and there was very less consensus between all three tools. We also added the citation to this paper in our manuscript:

“In other studies (Zufferey et al. 2018, Lazaris et al. 2017), similar disagreement between computational tools for TAD prediction was observed. It was also argued the parameter selection, normalization and matrix correction can have a significant impact of the final result of these tools (Lazaris et al 2017).”

Also, what was the bin size used for TAD calling? Using a 5kb resolution may introduce unnecessary noise, and I suggest repeating the analysis at a "standard" 40kb resolution.

Response: We have previously used 5kbp resolution data in our comparison experiment. As suggested by the reviewer, we repeated the experiment using the 40kbp resolution dataset of GM12878 of Bing Ren 's lab (Schmitt et al, Cell reports 2016). We obtained a similar result (figure below) as what we got from 5kbp resolution data. Again, the TAD fusion prediction varied a lots by using different TAD caller tools and there was very less consensus between all three tools.

More importantly, precisely because of issues with TAD callers in general, as pointed out by the authors and several studies, it is problematic to use them as a comparison to the proposed method.

Response: As the reviewer rightfully mentioned there were several issues with TAD caller algorithms, we moved this comparison result to the supplemental data. The purpose of this comparison now simply confirms that the TAD fusion prediction by using TAD callers can vary a lots from using different TAD caller tools.

Instead, the authors can use the concept of insulation score, as computed for each bin (not just at TAD boundaries). Then, their evaluation will not be dependent on whether the TAD boundary was called or not. The hypothesis here is that the higher the maximum insulation score inside the deletion region, the higher the probability of fusion. Therefore, this score should correlate with the proposed TAD fusion score.

Response: As the reviewer suggested, we compared our TAD-fusion score with the insulation score that was estimated at each bin using the newest tool LRI (<https://www.ncbi.nlm.nih.gov/pubmed/30184171>). First, we did observe a significant yet moderate correlation between the TAD fusion score assigned to each deletion and the highest LRI score assigned to any bin inside each deletion ($r = 0.43$, $p < 1e-10$). Next, we investigated and realized that the discordance between calculated TAD fusion score and the maximum insulation inside the deletions came from the cases where there were more than one strong insulator near the boundary. In these cases, some of insulators were not deleted/altered as a result of the deletion (see three figures attached sequentially below, the first two figures were from 5kbp resolution data, the third figure was from 40kbp resolution data). Thus the maximum insulation score may not represent exactly the TAD fusion probability.

Alternatively, we can use the insulation score summation of all loci of a deletion. We observed that the correlation between our TAD fusion score and LRI sum was stronger ($r = 0.5$, $p < 1e-14$). However, this method is still not exact since although the sum is large, it does not guarantee the deletion removes all insulators, there may still be some adjacent insulators not deleted.

Our method avoids this limitation by taking into account both the deleted insulators and the remaining (i.e. not deleted) insulators rather than considering only deleted ones (e.g. max/sum of them as above). Certainly, we can apply some heuristics on the insulation score to take into account both deleted and non-deleted insulators. For example, we can predict two loci will interact if the sum of remaining insulators is greater than a threshold. However, this method is too heuristic and not interpretable since we use two irrelevant heuristics/formulae: One is used to estimate the insulation from the interaction (i.e. Hi-C data) and another one is used to predict the interaction from the insulation. In our approach, we used an unique model for both estimating the insulation from the interaction and then predicting the interaction from the insulation.

We added the figure below into the supplemental data (Supplemental figure S6) and updated the manuscript to clarify that point (lines 162 - 171). The manuscript was updated as follows:

“The second approach we utilized for comparison was to use the maximum LRI score (i.e., an insulation score predicted by (Chen et al. NAR 2018)) of the deleted bins to rank/score each deletion. We did observe a significant yet moderate correlation between the TAD fusion score assigned to each deletion and the maximum LRI score assigned to any bin inside each deletion

($r = 0.43$, $p < 1e-10$, for chromosome 1). Further investigation has revealed that some of the discordance between calculated TAD fusion score and the maximum insulation score (LRI) inside the deletions came from the cases where there were more than one strong insulator near the TAD boundary. In these cases, some of insulators were not deleted/alterd as a result of the deletion (see Supplementary Figure S6). Thus, the deletion of a bin with high insulation score might not result in TAD fusion in these cases. Our method avoided this limitation by taking into account both the deleted insulators and the remaining (i.e. not deleted) insulators rather than considering only deleted ones.”

An illustrated example (one of 1KG deletions at chromosome 1) shows the limitation of the scoring method that only uses the maximum insulation score of bins inside the deletion. In

that case, this method gives a high score since the deletion removes a bin with a large insulation score (measured by LRI score or our insulation). Our method gives a low score for this deletion since there are still other insulators remaining (both the upstream and downstream of the deletion). The upper image is visualized with Juicebox, the lower image visualizes the LRI score and our insulation that are zoomed in around the deletion. Another example that is similar to the example above.

Another example but with 40kbp Hi-C data from Bing Ren 's lab.

3. Length of deletions

Some deletions in cancer can be really big, megabases long. Have the authors applied any upper limit? Figure 4d suggests so, but I am not entire sure if this is explicitly mentioned in the paper.

Response: We thank the reviewer for this comment. We have indeed applied an upper limit of 500kbp on our analysis of TCGA data precisely to avoid bias of very long deletions on the results as suggested by the reviewer. We have now updated the text to clarify this point (lines 221 - 222) as

“We limited the deletions from TCGA data to an upper bound of 500kbp to avoid bias of very long deletions”.

Also, in Figure 4d, there is a puzzling association of TAD fusion score with the size of the deletion. The authors need to explain why this is real, or, alternatively, look into their scoring approach for ways to normalize for the size.

Response: As the reviewer pointed out, the correlation between the TAD-fusion score and the length is expected since longer deletions will (i) reduce the genomic distance so that two distal loci become closer and thus have a higher chance of increasing the strength of interaction (our model takes into account the genomic distance between two bins to estimate their interaction), (ii) have a higher chance of deleting a TAD boundary and fusing two adjacent TADs. Furthermore, to make *sure the observed biological results is not due to length bias we have made sure to compare deletions of similar length range in all of the experiments as explained below.*

For example, I am not sure whether normalization over the size of the "rectangle" used for scoring has been done. After, this normalization, Figure 4c will probably be affected and this may change their conclusions.

Response: For all deletions (GreatApe, 1KG, TCGA), the TAD fusion score was calculated from the change of interactions between 500kbp upstream region and 500kbp downstream region of the deletion. Thus the size of the “rectangle” used for scoring was the same for all deletions (i.e. 500kbp x 500kbp) so the normalization of this size was not necessary.

We also have not applied any normalization of the *deletion length* in our score. Instead, in the analysis, we only made the score comparison between deletions with comparable length ranges. We merged figure 4c and 4d together (now become figure 4c) so that comparison of TAD fusion score on deletions of GreatApe, 1KG and TCGA was in the same length range (new Figure 4c is attached below). We can see that the TAD fusion score followed the order GreatApe < 1KG < TCGA in most of the cases of deletion lengths considered. In addition, a simple normalization (divide score by the deletion length) also gave the mean normalized score of GreatApe, 1KG, and TCGA of 0.006, 0.0071, and 0.0073 respectively (GreatApe vs 1KG: p-value < 1e-4, 1KG vs TCGA: p-value < 1e-2) which agreed with the results presented. Thus

our overall conclusion was valid and not changed. We updated the figure 4 and the main text to clarify these points (lines 220 - 233). The manuscript is updated as follow:

“We also compared the TAD-fusion score for fixed deletions in great ape lineages (Sudmant et al. GR 2013), 1KG deletions and TCGA (cancer patients) deletions of the same length ranges. We limited the deletions from TCGA data to an upper bound of 500kbp to avoid bias of very long deletions.”

Other comments:

a. Lines 26-27: These contacts are still between bin pairs of variable distance, so HiCRep should be used here as well instead of regular Pearson correlation.

Response: We replaced the Pearson correlation by the SCC score of HiCRep as the reviewer suggested. We changed the text as *“Since these interactions were at variable distance, we used the stratum adjusted correlation coefficient (SCC) from HiCRep package (Yang et al, 2017) to evaluate the similarity instead of regular Pearson correlation coefficient.”*

b. Line 34: Armatus is misspelled

Response: We corrected it.

c. Figure S1, clearly mark the rectangle of where that TAD fusion score is calculated on the Hi-C heatmaps

Response: We marked the region that we calculated the TAD fusion score in figure S1 (now became figure S3).

Introduction:

a. Line 44-46: Reference [24], in agreement with Lazaris et al, BMC Genomics 2017, shows that some TAD callers are quite consistent, whereas calling specific loops is a much harder problem. Therefore, the authors should appropriately rephrase.

Response: We rephrased as *“Although the number and the size of TADs vary significantly among different tools, this result is still more comparable than the one from the problem of determining specific loops (Lazaris et al 2017, Forcato et al 2017).”*

b. Line 47-48: Other reasons for lack of high concordance can be sequencing depth, and differences in library preparation (e.g. different cutting enzymes)

Response: We added this in our text as *“In addition, the difference in sequencing depth and library preparation (e.g. different cutting enzymes) also contributes to this discordance.”*

c. Line 59: the disruption in the PDGFRA locus is due to methylation, not deletion (reference [42]).

Response: We thank the reviewer for this correction, we removed this example from the introduction.

d. Two more papers to cite: [https://www.cell.com/cell/fulltext/S0092-8674\(18\)31021-3](https://www.cell.com/cell/fulltext/S0092-8674(18)31021-3) and <https://www.ncbi.nlm.nih.gov/pubmed/27869826>

Response: We thank the reviewer for pointing out these papers. We have cited them in the manuscript now (lines 53 - 54, 57 - 58).

e. Negative selection of deletions of TAD boundaries in a pan-cancer analysis has been recently demonstrated in <https://www.ncbi.nlm.nih.gov/pubmed/29416042>. Actually, this paper may be useful for computing the insulation scores (proposed in my comments above), as it shows a strong association between insulation score, CTCF binding strength and negative selection of deletions

Response: We are thankful for this suggestion. We have now cited this paper and re-stated their result as

“A recent study reported the negative selection of deletions of TAD boundaries and its effect on pan-cancer genome analysis (Gong et al, 2018). Furthermore, they showed that not only very strong TAD boundaries were protected from deletion, but they tended to be co-duplicated with super-enhancer (Gong et al., 2018). In this study, we explored further the TAD-fusion score for deletions reported in tumor cells from TCGA (The Cancer Genome Atlas) by comparing them against deletions from the 1000 genomes project of similar length.”

We tried using the HiC-bench as the reviewer suggested. Unfortunately, we could not install this toolbox on our cluster since one of the required packages (genlasso) was removed from CRAN in the new R version 3.5 (HiC-bench used genlasso from CRAN 3.0). However, we did find LRI (Chen et al. NAR 2018), the most recently published tool, and we realized that LRI was very similar to the hic-ratio score (of HiC-bench) as both of them normalized the inter-TAD interaction by the intra-TAD interaction. Thus, we have used the LRI score in most of our comparisons in the paper.